# Clinical and Laboratory Associations with Methotrexate Metabolism Gene Polymorphisms in Rheumatoid Arthritis

**DOI:** 10.3390/jpm10040149

**Published:** 2020-09-26

**Authors:** Leon G. D’Cruz, Kevin G. McEleney, Kyle B. C. Tan, Priyank Shukla, Philip V. Gardiner, Patricia Connolly, Caroline Conway, Diego Cobice, David S. Gibson

**Affiliations:** 1Northern Ireland Centre for Stratified Medicine (NICSM), Biomedical Sciences Research Institute, Ulster University, C-TRIC Building, Londonderry BT47 6SB, UK; darthcruz@gmail.com (L.G.D.); kmceleney29@gmail.com (K.G.M.); Tan-BC@ulster.ac.uk (K.B.C.T.); p.shukla@ulster.ac.uk (P.S.); 2Respiratory Medicine Department and Clinical Trials Unit, Queen Alexandra Hospital, Portsmouth PO6 3LY, UK; 3Rheumatology Department, Western Health and Social Care Trust, Londonderry BT47 6SB, UK; Philip.Gardiner@westerntrust.hscni.net; 4Cardiac Assessment Unit, Western Health and Social Care Trust, Omagh BT79 0NR, UK; Patricia.Connolly@westerntrust.hscni.net; 5Mass Spectrometry Centre, Biomedical Sciences Research Institute (BMSRI), School of Biomedical Sciences, Ulster University, Cromore Road, Coleraine BT52 1SA, UK; c.conway@ulster.ac.uk (C.C.); d.cobice@ulster.ac.uk (D.C.)

**Keywords:** rheumatoid arthritis, SNP, DMARD, methotrexate, pharmacogenomics

## Abstract

Rheumatoid arthritis (RA) is a chronic systemic autoimmune disease that causes loss of joint function and significantly reduces quality of life. Plasma metabolite concentrations of disease-modifying anti-rheumatic drugs (DMARDs) can influence treatment efficacy and toxicity. This study explored the relationship between DMARD-metabolising gene variants and plasma metabolite levels in RA patients. DMARD metabolite concentrations were determined by tandem mass-spectrometry in plasma samples from 100 RA patients with actively flaring disease collected at two intervals. Taqman probes were used to discriminate single-nucleotide polymorphism (SNP) genotypes in cohort genomic DNA: rs246240 (*ABCC1*), rs1476413 (*MTHFR*), rs2231142 (*ABCG2*), rs3740065 (*ABCC2*), rs4149081 (*SLCO1B1*), rs4846051 (*MTHFR*), rs10280623 (*ABCB1*), rs16853826 (*ATIC*), rs17421511 (*MTHFR*) and rs717620 (*ABCC2*). Mean plasma concentrations of methotrexate (MTX) and MTX-7-OH metabolites were higher (*p* < 0.05) at baseline in rs4149081 GA genotype patients. Patients with rs1476413 SNP TT or CT alleles have significantly higher (*p* < 0.001) plasma poly-glutamate metabolites at both study time points and correspondingly elevated disease activity scores. Patients with the rs17421511 SNP AA allele reported significantly lower pain scores (*p* < 0.05) at both study intervals. Genotyping strategies could help prioritise treatments to RA patients most likely to gain clinical benefit whilst minimizing toxicity.

## 1. Introduction

Rheumatoid arthritis (RA) is the most common chronic autoimmune inflammatory arthritis, affecting approximately 0.3–1% of the world’s population [1,2]. The disease primarily affects the articular joints, causing swelling, stiffness, joint destruction [3], loss of function in joints [4], disability and a significantly lower quality of life. To prevent irreversible joint damage resulting in substantial disability, it is important to introduce disease-modifying anti-rheumatic drugs (DMARDs) early after onset and failure of non-steroidal anti-inflammatory treatment.

Conventional synthetic disease-modifying anti-rheumatic drugs (csDMARDs) such as methotrexate (MTX), hydroxychloroquine (HCQ), cyclosporin, sulfasalazine (SSZ) and leflunomide are commonly used mainstays of the disease; however, it is widely known that a significant proportion of patients with RA often show poor or inadequate therapeutic response to csDMARDs [5]. The anti-folate MTX is the cheapest drug in treatment of RA and is often the first-line treatment [6]; however, only 55% of patients remain on this drug for more than 2 years due to a build-up of non-response or the accumulation of various adverse side effects [6,7]. MTX is subject to significant metabolic activity in the body; the polyglutamated derivatives of MTX are selectively retained in cells, therefore lengthening the activity of the drug which complicates treatment management, since patients would continue taking their daily drug dosage oblivious to the fact that their circulating drug levels are still high, potentially contributing to undesirable cytotoxic effects [8,9]. MTX is converted in hepatic parenchymal cells resulting in the 2- through 4-glutamate residues derivatives or the drug is catabolised to the 7-hydroxy-methotrexate (MTX-7-OH) form [10]. More than 10% of a dose of methotrexate is oxidised to MTX-7-OH, irrespective of the route of administration [11]. The MTX-7-OH metabolite is extensively (91 to 93%) bound to plasma proteins, in contrast to the parent drug (only 35 to 50% bound) and contributes to inactivity of the drug or poor response to treatment [11].

When non-response has been confirmed, NICE clinical guidelines recommend switching to the more costly biological disease-modifying anti-rheumatic drugs (bDMARDs) [5,12,13]. Various studies indicate that treat-to-target strategies which aim to reduce disease activity shortly after diagnosis result in better long term outcomes and can minimise permanent joint damage, thus there is a genuine need for earlier identification of patients who do not respond well to csDMARDs treatments [6].

It is estimated that 15–30% of variation in drug responses are attributable to genetic or single-nucleotide polymorphisms or SNPs [14]. Not all SNPs are functional; some are in non-coding areas (introns) and there is a variety of ways that a SNP can affect or inhibit downstream transcription factor, gene or protein function [15]. The promise of pharmacogenomics is that identification of SNPs and associated risk alleles could identify patients who may be susceptible to accumulating cytotoxic levels of a drug during therapy (as in the polyglutamate derivatives of MTX) or when certain metabolite levels accumulate rendering a drug as inactive (as in MTX-7-OH).

In this study, we sought to determine the metabolite levels in RA patients taking DMARDs. We then carried out genotyping of 10 SNPs known to influence the metabolic pathways of DMARDs in arthritis. Our aim was to determine if genetic variations or polymorphisms associate with metabolite levels. This could help design studies to improve clinical management, which risk stratify patients at greater predisposition of forming ineffective or potentially harmful metabolite levels, by adequately planning ahead the appropriate drug and dosage.

## 2. Materials and Methods

### 2.1. Participant Recruitment

The research team at Ulster University collaborated with rheumatologists from the Western Health and Social Care Trust (WHSCT) to design, conduct and recruit patients to the study. Informed consent to participate was obtained from all RA patients enrolled to the study. One hundred patients identified using following inclusion/exclusion criteria were recruited into the prospective observational cohort study: Remote Arthritis Disease Activity MonitoR (RADAR); ClinicalTrials.gov Identifier: NCT02809547. Inclusion criteria: aged between 18–90 years, diagnosed with RA (according to American College of Rheumatology criteria [5,16]), diagnosed with RA for a minimum of 1 year and maximum 10 year duration, active disease flares on a regular basis, and receiving a disease-modifying anti-rheumatic drug (DMARD). Exclusion criteria: any other inflammatory conditions, any infections or trauma during study period, and have restricted hand function (determined by clinical team). Office for Research Ethics Committees Northern Ireland (ORECNI) (16/NI/0039), Ulster University Research Ethics Committee (UREC) (REC/16/0019) and WHSCT (WT/14/27) approvals were obtained for the study. Informed consent was obtained from all RA patients enrolled to the study. 

### 2.2. Whole Blood and Dried Blood Spot Sample Collection

Venepuncture whole blood samples as part of the normal routine care pathway were forwarded to the hospital laboratories for multiple tests including CRP, ESR, Bilirubin, Liver enzymes and full blood count. An additional 5-mL EDTA tube of blood was collected from each of the 100 participants for DNA genotype and drug metabolite analyses within the RADAR study. Samples were collected at both study baseline and at a 6-week follow-up appointment at an outpatient rheumatology clinic for all participants. Additionally, a sub cohort of 30 of the above participants were supplied with a kit containing sufficient dried blood spot (DBS) cards [17], finger lancets and pre-paid and addressed postal envelopes with desiccant and biohazard sealable pouch to send a weekly samples (approximately 3–5 droplets of blood ~20 μL each) from home to the NICSM laboratory for drug metabolite analysis. Finger lancet blood droplets were deposited onto dried blood spot (DBS) Protein Saver 903TM cards (Whatman, GE Healthcare Life Sciences, Buckinghamshire, UK), pre-treated with a protein stabiliser coating.

### 2.3. Nucleic Acid Isolation from Peripheral Blood

Total DNA was isolated from peripheral blood samples using TRIzol reagent (TRIzol LS Reagent, Thermo Fisher Scientific, Basingstoke, UK. cat. No 10296-010) according to manufacturer’s directions. Total DNA concentration was estimated by spectrophotometry (NanoVue Plus—GE Health Care, Buckinghamshire, UK).

### 2.4. Mass-Spectrometry Analysis

Determination of methotrexate (MTX) metabolites was performed using a liquid extraction surface analysis (LESA) coupled with nanoESI-triple quadruple mass spectrometer (QQQ) using Triversa nanomate (Advion, New York, NY, USA) and API 4000 QQQ Mass Spectrometer (AB Sciex, Cheshire, UK). Control MTX metabolites and internal standards were from Schircks Laboratories (Jona, Switzerland).

Quantitation for MTX and MTX metabolites was performed by the matrix-matched standards approach using an intensity ratio (ISTD/MTXs) calibration (10–2000 nM). Signal for each metabolite was the average of *n* = 2 (duplicate injection). A total of 5 nM was selected as LLOD (S/N ~ 3) for MTX, 7-OH MTX and MTX-PG2 and 8 nM was selected for MTX-PG3 to PG5; 10 nM was selected as LLOQ for MTX and all metabolites (S/N ~ 10). Intra- and inter-day precision was assessed at both 50 and 500 nM and coefficient of variation (CV) for MTX metabolites ranged from 2.0–7.2%. Linear regression coefficient (R2) of the back-calculated concentration against the nominal concentration for MTX and its metabolites was above 0.995.

Determination of sulfasalazine metabolites and teriflunomide, analysis was performed by liquid chromatography tandem mass spectrometry (LC-MS/MS) using a HP 1200 LC (Agilent, Palo Alto, CA, USA) and a Quattro micro mass spectrometer (Micromass, Manchester, UK). Control sulfasalazine metabolites, teriflunomide and internal standard were obtained from Sigma Aldrich, Gillingham, UK. Quantitation of sulfasalazine, its metabolites and teriflunomide was performed by the matrix-matched standards approach using an intensity ratio (ISTD/Analyte) calibration (5–500 µg/L). Signal for each metabolite was the average of *n* = 2 (duplicate injection). 5 µg/L was selected as LLOD (S/N > 5) for metabolites and 10 µg/L was selected as LLOQ for all metabolites (S/N > 10).

Intra- and inter-day precision was assessed at both 20 and 100 µg/L and coefficient of variation (CV) for sulfasalazine metabolites ranged between 1.4–5.8%. Linear regression coefficient (R2) of the back-calculated concentration against the nominal concentration for sulfasalazine, its metabolites and teriflunomide was above 0.992.

Separation of targeted analytes was carried out by reverse phase chromatography using a C18 column in gradient mode. Quantitation of all analytes were performed in positive ion mode multiple reaction monitoring (MRM) using matrix-matched standards and stable isotope ratios. All Mass Spectrometry methods were validated according to ICH Guidance for selectivity/specificity, limit of detection/quantitation (LLOD/LLOQ), linearity and precision [18]

### 2.5. Endpoint-Genotyping Using Taqman Assay

Endpoint genotyping analysis was carried out using the LightCycler 480 real-time PCR system (ROCHE). The assay is based on the competition during annealing between probes detecting the wild type and the mutant allele. The 5′-exonuclease activity of DNA polymerase cleaves the doubly labelled Taqman probe hybridised to the SNP-containing sequence, once cleaved, the 5′-fluorophore is separated from a 3′-quencher. Two allele-specific probes carrying different fluorophores (VIC^®^, emission: 554 nm and FAMTM, emission: 518 nm) permits SNP determination in a single well without any post-PCR processing. Genotype is determined from the ratio of intensities of the two fluorescent probes at the end of amplification (endpoint instead of the entire cycle in conventional PCR).

### 2.6. Taqman Probes Used for Single-Nucleotide Polymorphism Genotyping

The concentration and integrity of the genomic DNA were assessed by microvolume spectrophotometer (NanoDrop 2000; Thermo Fisher Scientific). DNA samples were genotyped by the following TaqMan SNP genotyping assays [*MTHFR*-rs1476413, Assay Identification (ID) C_8861304_10; RS1-rs 2231142, Assay Identification (ID) C_354526997_10; *ABCC2*-rs3740065, C_22271640_10; *SLCO1B1*-rs4149081, Assay Identification (ID) C_1901759_20; *MTHFR*-rs4846051, Assay Identification (ID) C_25763411_10; *ABCB1*-rs10280623, Assay Identification (ID) C_30537012_10; *ATIC*-rs16853826, C_33295728_10; *MTHFR*-rs17421511, Assay Identification (ID) C_32800189_20; *ABCC2*-rs717620, C_2814642_10; *ABCC1*-rs246240, Assay Identification (ID) C_1003698_10; Life Technologies Ltd.).

### 2.7. Validation of Polymorphisms by Pyrosequencing

Validation of SNP genotyping results from the Taqman assays was performed on a subset of samples using pyrosequencing. Due to the high number of SNPs to cover, a method using a universal biotinylated primer was employed [19]. Briefly, this method involves the use of standard target specific primer pairs with a universal M13 sequence at the 5′ end of one of the primers. A third, biotinylated M13-targeting primer is included in the PCR amplification reaction, leading to incorporation of biotin into the PCR product without the need for individual biotin labelling of each individual primer pair and thus lowering the cost of pyrosequencing considerably. A list of primers used for pyrosequencing are shown in Appendix A.

PyroMark Assay Design Software 2.0 (Qiagen) was used for primer design in the SNP calling assay design format. PCR amplification was carried out using the Pyromark PCR kit (Qiagen) in 25 μL total volumes with 10–20 ng DNA and final concentrations of 0.2 mM for each primer. Standard PCR cycling conditions were used as per manufacturer’s instructions and were consistent for all samples and targets. PCR products were checked by agarose gel electrophoresis and those with a positive single band of the expected size were taken forward into pyrosequencing on the Pyromark Q48 (Qiagen) using standard manufacturers protocols and the instrument run setting for SNP calling.

### 2.8. Statistics and Sample Size Calculations

Statistical analysis was carried out with SPSS ver.25 (IBM Corp, NY, USA), SciPy module (ver. 1.3) for Python (version 3.7.2) and R (version 3.60) with *p* < 0.05 considered as statistically significant, all within 95% confidence intervals. Descriptive statistics were used to characterise the variability in mean MTX, MTX-7-OH and MTX2PG–5PG and MTXtotal concentrations between different genotype groups of patients. Normality of data was determined using the Shapiro-Wilks test in SPSS (ver. 25) prior to employing the Kruskal–Wallis non-parametric (distribution free) one-way ANOVA, with Dunn–Bonferroni post hoc test to assess differences between genotype group means using GraphPad Prism version 8.0.0 for Windows (GraphPad Software, San Diego, CA, USA, www.graphpad.com). 

The Hardy-Weinberg equilibrium was assessed for SNPs with significant clinical associations in the methotrexate treated cohort. A Chi-square test with Benjamini-Hochberg adjusted *p* values was used to assess if there were significant differences between the genotype frequencies expected from dbGAP European population and those observed. Power was calculated for the same SNPs using the GENPWR package [20] within R (v 4.0.2), using the linear regression model (with alpha at 0.05) since the goal was to calculate power in a continuous outcome (metabolite levels) between genotypes.

## 3. Results

### 3.1. Study Population

A total of 100 participants, *n* = 68 female and *n* = 32 male, with active rheumatoid arthritis were enrolled to this study (Table 1). The mean age of study participants was 59.5 years with a mean disease duration of 6 years and mean baseline disease activity score (DAS28ESR) of 3.6.

A subgroup of *n* = 66 participants (*n* = 46 female) were treated with weekly methotrexate at baseline was identified for subsequent genotype association analyses. Baseline and 6-week follow-up drug dose information is summarized in Appendix A for this main subgroup. A smaller, partially overlapping, subgroup of *n* = 27 participants (*n* = 20 female) being treated with daily sulfasalazine at baseline was also identified for subsequent analyses (Appendix A). 

### 3.2. Single-Nucleotide Polymorphisms Analysed in this Study

Ten SNPs were analysed in this study, previous studies have linked these SNPs to various clinical consequences observed in RA patients being treated with DMARDs, documented in the PharmGKB database [21,22]. SNP genotypes were determined by endpoint PCR assay and allele specific probes (Figure 1). The SNPs characteristics and frequencies in the study cohort are summarised in Table 2.

### 3.3. Methotrexate and Sulfasalazine Metabolite Polymorphism Associations

The data strongly suggest plasma concentrations of methotrexate and sulfasalzine metabolites are associated with the allelic genotype for 2 particular polymorphisms, rs4149081 and rs1476413 (see Figure 2). Table 3 indicates that within the *n* = 66 methotrexate treated subgroup, *n* = 17 participants with the minor homozygote genotype AA in rs4149081 have a significantly lower mean plasma MTX-7-OH concentration compared to the GA genotype group (*p* = 0.002) at baseline. Although a similar trend is observed at the 6-week follow-up, this was not statistically significant. The GA genotype group mean concentrations of MTX and MTX-7-OH are significantly higher than those observed in the GG genotype group (*p* = 0.01 and *p* = 0.038, respectively; Figure 2A,B). The baseline mean blood bilirubin concentration was the only feature observed at significantly higher levels in the rs4149081 AA genotype, relative to the GG genotype group (*p* = 0.020; Figure 2C). 

A total of *n* = 8 participants with the rs1476413 homozygous major allele genotype CC have significantly lower (*p* = 0.012) group mean plasma MTX-7-OH concentration, compared to the CT genotype group (Table 3) at the 6-week follow-up sessions. No significant difference was observed at baseline. The mean plasma concentration of tetraglutamate MTX metabolites are also significantly lower in the rs1476413 CC genotype group at both baseline (*p* = 0.02) and 6-week follow-up appointments (*p* = 0.008; Figure 2F,G and Table 3).

A total of n = 6 participants with the minor allele genotype AA in the SNP rs17421511 (Appendix A) show significantly higher mean plasma concentration (*p* = 0.013) of sulfapyridine at the 6-week follow-up appointment period only, relative to GG and GA genotype groups.

### 3.4. Clinical and Laboratory Feature Polymorphism Associations

While the average red blood cell counts (RBC) remain within reference ranges in both men and women in this study (Table 1), there is a modest but statistically significant decrease in mean RBC levels in the *n* = 15 participants with the rs2231142 heterozygous genotype GT compared to the TT genotype group at both study time points (*p* = 0.044; Table 3, Figure 3A,B). Mean alkaline phosphatase concentration is also significantly lower at baseline in the rs2231142 GT genotype group, relative to the TT genotype participants (*p* = 0.019; Figure 3C). In the smaller group of *n* = 6 rs2231142 GG genotype participants, mean lymphocyte counts are significantly higher than the GT genotype group, again at both time points (*p* = 0.043, *p* = 0.019; Figure 3D,E).

Mean patient-reported general pain (PgPain) scores are significantly lower in *n* = 9 participants carrying the rs17421511 AA genotype at baseline (*p* = 0.033) and at the 6-week follow-up appointment (*p* = 0.013) compared to those with the GA genotype (Figure 3F,G). This trend is also reflected in significantly lower mean baseline DAS28ESR scores in participants carrying the rs17421511 AA genotype, relative to the GA and GG genotype groups (*p* = 0.002 and *p* = 0.005 respectively; Table 3, Figure 3H), though scores even-out at the 6-week follow-up period among all three genotypes. The baseline mean alanine aminotransferase (ALT) was recorded at significantly higher blood concentrations in the rs17421511 AA genotype group, relative to the GG genotype participants (*p* = 0.048; Figure 3I).

The Benjamini–Hochberg-adjusted Chi-square test p-values showed no statistically significant differences between the genotype frequencies observed and those expected from dbGAP European population: rs4149081, p*_adj_* = 0.515; rs 1476413, p*_adj_* = 0.945; rs17421511, p*_adj_* = 0.711; rs2231142, p*_adj_* = 0.9454. The power calculated for each of these SNPs at *p* < 0.05 was: rs4149081, 0.96; rs 1476413, 0.95; rs17421511, 0.97; rs2231142, 0.94.

## 4. Discussion

This study investigates the influence of ten well-characterised SNPs in RA and we have tried to correlate this with the appearance and accumulation of metabolites measured in the plasma of patients taking DMARDs such as methotrexate and or sulfasalazine. The in vivo pharmacotherapy of DMARDS and potential response biomarkers in RA have been previously described [6,23,24,25], however there studies of potential associations between circulating csDMARD levels and specific genetic variants remain limited in RA patients.

Typically, methotrexate treatment may cause elevations in serum AST and ALT, long term therapy has also been linked to development of fatty liver disease, fibrosis, cirrhosis, nephrotoxicity, and renal failure [26]. However, under active consultant-led clinical management, these effects are largely minimised. The mean values of all clinical biomarkers, liver enzyme and blood component cell-counts (Table 1) are within recommended normal reference ranges when viewed across all of the study participants. However, mean ALT was significantly higher at baseline in the methotrexate treated subgroup of participants with the rs17421511 AA genotype, albeit the potential effect of multiple drug combinations was not investigated in this subgroup.

Although the average RBC count when taken from all participants appear to be within normal ranges (Table 1), participants with the GT allele in the rs2231142 SNP have significantly decreased erythrocyte counts in their circulation compared to those with the homozygous alleles methotrexate affects folic acid metabolism, thus patients taking MTX may show variations in their mean corpuscular volume (MCV) of red blood cells (RBC), therefore resulting in megaloblastic anaemia.

RBCs retain MTX as the polyglutamate derivatives throughout their lifespan [27,28]. While normal RBC levels are between 4.7 to 6.1 million cells per microlitre (mill.c/µL) for men and between 4.2 to 5.4 mill.c/µL in women, the slight decrease shown in heterozygous rs2231142 SNP patients is statistically significant. However, the lower mean haemoglobin levels observed in the rs2231142 GT genotype is not statistically significant and there is no correlation with disease activity score as may have been anticipated in anaemia of chronic disease.

Apart from the impact of sex-linked genes in RA, the diversity in our genomes are partially accountable for the heterogeneity in the clinical presentation of synovitis among patients [29]. The genetic influence in RA is particularly strong, the heritability in RA is estimated to be around 60% [29] and with the high diversity of clinical presentations observed in RA, the goal in treatment would be to stratify patients according to their genetic profile and clinical outcome, eventually formulating a genetic-risk based personalised treatment management protocol.

MTX is an anti-folate drug, with anti-proliferative and anti-inflammatory effects, by inhibition of folate and adenosine pathways and inhibition of purines and pyrimidines synthesis [30,31,32]. Approximately 80–90% of methotrexate is primarily excreted by the kidneys [33]. MTX is converted in hepatic parenchymal cells of some patients resulting in the 2- through 4-glutamate residues derivatives or the drug is catabolised to the MTX-7-OH form. 

Though not observed consistently on both study time points, participants with the AA allele in rs4149081 and CC allele in rs1476413 can have significantly lower mean plasma levels of MTX-7-OH in their plasma circulation. Since some genotype groups are modest in size, the potential for differences in the mean MTX dose between genotype groups was analysed, though no significant difference was observed (Appendix A). Furthermore, only a weak correlation exists between MTX dose and circulating MTX-7-OH (r2 = 0.08213). Increasing levels of MTX-7-OH is known to inhibit the clinical responsiveness of RA patients to the MTX drug and therefore, reduced levels of this metabolite could signify a better clinical response to MTX [34]. Thus, with genetic profiling of expanded csDMARD naïve RA cohorts, it would be interesting to further investigate clinical responsiveness to MTX in these genotypes.

The 2- through 4-polyglutamate MTX metabolite-derivatives are selectively retained in cells and participants with the TT or CT alleles in the rs1476413 SNP tend to show significantly higher mean plasma levels of the tetraglutamate metabolite. A significantly higher mean DAS28 is observed only at study baseline in the rs1476413 CT genotype group relative to CC genotype, though due to modest numbers in the latter group this would require independent verification. It is likely that folic acid supplementation in the study cohort to mitigate toxicity of MTX has reduced the frequency of observable side effects.

Sulfasalazine is metabolized by intestinal bacteria, resulting in the release of sulfapyridine (SPY) and 5-aminosalicylate or 5-ASA (SPY and 5ASA are linked by an azo bond) [35]. Sulfapyridine is almost completely absorbed by the colon, metabolized by the liver, and renally excreted [1]. Commonly reported side-effects of sulfapyridine are minor gastrointestinal (GI) and central nervous system (CNS) abnormalities, and uncommon serious haematological and hepatic side-effects [36,37]. Although study participants with the AA and GA alleles of the rs17421511 SNP indicate higher mean plasma levels of sulfapyridine compared to those with the GG allele (Appendix A), no significant adverse phenotypic effects were observed in these subgroups.

The modest number of patients with the AA genotype of the rs17421511 SNP in our study report significantly lower levels of pain and disease activity, relative to the remaining methotrexate treated cohort. In future research with expanded patient cohorts, it would be pertinent to see if this phenomenon is observed in other patient groups carrying this particular genotype. As a general observation, a limitation of the current study is the low number of participants in particular genotype groups and the smoking status was not recorded, which may impact upon methotrexate metabolism. Furthermore, the findings for the methotrexate treated cohort are only generalizable to the European population, as no significant differences in Hardy–Weinberg equilibrium were found by Chi-square test between the dbGAP frequencies and those observed in this study.

While it is challenging to find a clear-cut relationship between genotype and circulating drug levels which translates through to a clear prediction of phenotypic consequence, useful leads are presented in the current study. The rs1476413 and rs17421511 *MTHFR* variants and the rs2231142 *ABCG2* variant display significant changes which are consistent at both study time points. 

With further carefully powered studies of variability in both csDMARD response and predisposition to side effects, there is considerable potential to personalise effective treatments whilst avoiding any toxicity.

## Figures and Tables

**Figure 1 jpm-10-00149-f001:**
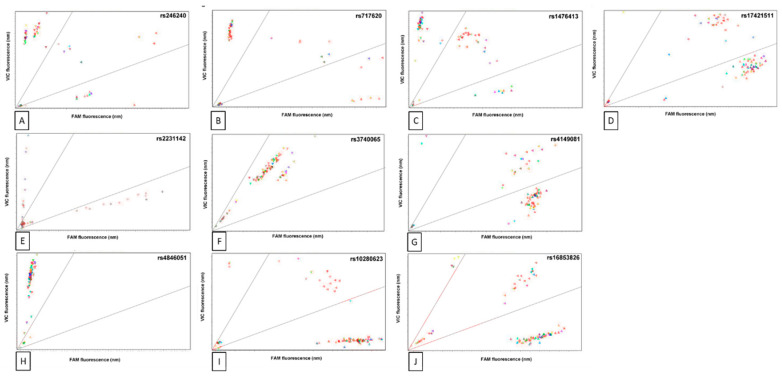
Endpoint polymorphism genotype assay. Genotype were determined for the listed polymorphisms in the RADAR study cohort (*n* = 100) from the ratio of fluorescent intensities (nm) of the two-allele specific Taqman probes (VIC and FAM) at the end of PCR amplification. Clusters in upper and lower quadrants represent groups of individuals with a homozygous genotype for either allele; the middle quadrant represents individuals with heterozygous genotype. (**A**) rs246240, (**B**) rs717620, (**C**) rs1476413, (**D**) rs17421511, (**E**) rs2231142. (**F**) rs3740065, (**G**) rs4149081, (**H**) rs4846051, (**I**) rs10280623, (**J**) rs16853826. Genotypes were also confirmed by pyrosequencing in selected individuals. Genotype frequency is summarized in Table 1.

**Figure 2 jpm-10-00149-f002:**
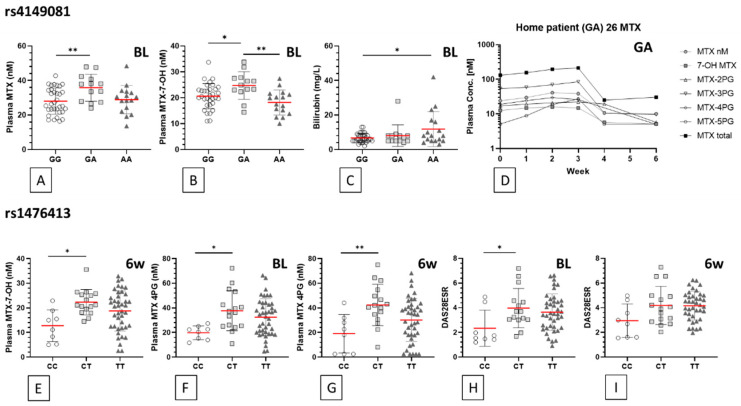
rs4149081 and rs1476413 genotype associations. Statistically significant associations between two polymorphisms, plasma drug metabolite concentration, other laboratory and clinical outcome measures are shown for individual genotypes in methotrexate treated participants (*n* = 66). Each symbol represents an individual participant of the genotype indicated on the x axes. Data grouped by rs4149081 genotypes: (**A**) baseline plasma concentration of unmetabolised methotrexate, (**B**) baseline plasma concentration of 7-hydroxy-methotrexate, (**C**) baseline bilirubin blood concentration, (**D**) weekly plasma concentrations (log scale) of listed methotrexate metabolites (PGs: polyglutamate subtypes) of a rs4149081 GA genotype participant. Data grouped by rs1476413 genotypes: (**E**) 6-week follow-up plasma concentration of 7-hydroxy-methotrexate, (**F**) baseline and (**G**) 6-week follow-up plasma concentrations of long-chain methotrexate 4-glutamate, (**H**) baseline and (**I**) 6-week follow-up disease activity (DAS28ESR) scores. Statistically significant differences between genotype group means are indicated by horizontal bars and an asterisk used to summarise *p* values adjusted by Bonferroni’s multiple comparison test: (*) *p* < 0.05; (**) *p* < 0.005 (descriptive statistics data shown in Table 3). Red horizontal bar represents genotype group mean; error bars represent standard deviation. MTX: methotrexate; BL: baseline; 6w: 6-week follow-up.

**Figure 3 jpm-10-00149-f003:**
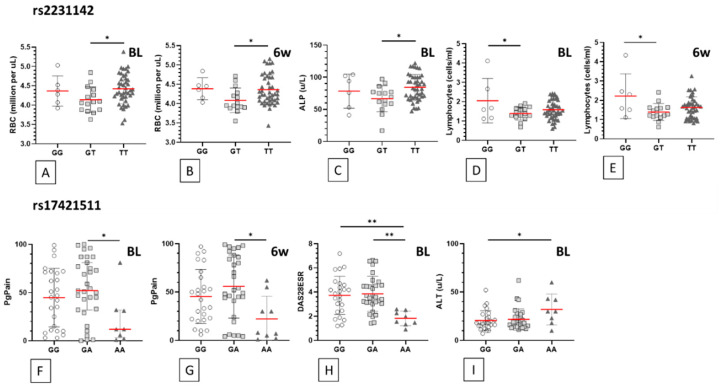
rs2231142 and rs17421511 genotype associations. Statistically significant associations between two polymorphisms and other laboratory and clinical outcome measures are shown for individual genotypes in methotrexate treated participants (*n* = 66). Each symbol represents an individual participant of the genotype indicated on the x axes. Data grouped by rs2231142 genotypes: (**A**) baseline and (**B**) 6-week follow-up red blood cell (RBC) count, (**C**) baseline blood alkaline phosphatase (ALP) concentration, (**D**) baseline and (**E**) 6-week follow-up lymphocyte count. Data grouped by rs17421511 genotypes: (**F**) baseline and (**G**) 6-week follow-up patient assessed pain (PgPain) levels, (**H**) baseline disease activity (DAS28ESR) scores, (**I**) baseline blood alanine aminotransferase (ALT) concentration. Statistically significant differences between genotype group means are indicated by horizontal bars and an asterisk used to summarise *p* values adjusted by Bonferroni’s multiple comparison test: (*) *p* < 0.05; (**) *p* < 0.005 (descriptive statistics data shown in Table 3). Red horizontal bar represents genotype group mean; error bars represent standard deviation. BL: baseline; 6w: 6 weeks follow-up.

**Table 1 jpm-10-00149-t001:** RADAR Study Cohort Demographics. Clinical and laboratory feature summary across *n* = 100 participants. yr: years; Anti-CCP: anti-cyclic citrullinated peptide; ESR: erythrocyte sedimentation rate; DAS28: disease activity score across 28 joints; RBC, red blood cell (count); Hb: haemoglobin; WBC: white blood cells; ALT: alanine aminotransferase; AST: aspartate aminotransferase; ALP: alkaline phosphatase; SD: Standard deviation.

	Mean	SD
Gender, male/female	32/68	-
Age, yr	59.5	12.6
Disease duration, yr	6.0	3.9
C-reactive protein, mg/L	7.9	11
Rheumatoid factor, n positive	63	-
Anti-CCP, n positive	53	-
ESR (mm/h)	15	15
DAS-28 ESR	3.6	1.5
RBC (cells/mL)	4.5	1.2
Hb (g/100 mL)	134	13
WBC (cells/mL)	6.8	2.3
Neutrophils (cells/mL)	4.4	1.9
Lymphocytes (cells/mL)	1.7	1.4
Platelets (cells/mL)	254	68
ALT (u/L)	23	11
AST (u/L)	24	9.4
ALP (u/L)	80	22
Bilirubin mg/L	8.2	5.6

**Table 2 jpm-10-00149-t002:** Characteristics of the single nucleotide polymorphisms analysed in the RADAR study cohort. Ref: reference; Alt: alternative; dbGAP: database of Genotypes and Phenotypes; Genotype frequency for *n* = 100 participants, −/− or +/+: homozygote; +/−: heterozygote.

					Minor Allele Frequency (European; dbGAP Sample Size)	Genotype Frequency		
SNP	Variant Locale	Gene	Clinical Consequences	Alleles (Ref > Alt)		−/−	+/−	+/+
rs10280623	Intron	ABCB1	Associated with MTX toxicity	T > C	C = 0.1933 (*n* = 5,216)	64	21	14
rs246240	Intron	ABCC1	Low response to MTX	A > G	G = 0.153169 (*n* = 131,156)	6	25	69
rs717620	Non-coding Transcript	ABCC2	Increased time needed to reach therapeutic level	C > T	T = 0.198186 (*n* = 137,144)	11	24	65
rs3740065	Intron	ABCC2	Increased risk of MTX toxicity	A > G	G = 0.108811 (*n* = 128,976)	1	10	87
rs2231142	Intron Missense	ABCG2	SNP causes higher toxicity with combination treatments	G > T	T = 0.10454 (*n* = 88,504)	8	21	71
rs16853826	Intron	ATIC	ATIC rs16853826 variant associated with toxicity	G > A	A = 0.12231 (*n* = 10,972)	72	20	8
rs4846051	Codon Synonymous	MTHFR	Increased risk of MTX toxicity	A > G	G = 0.023185 (*n* = 117,404)	0	0	100
rs17421511	Intron	MTHFR	Positive response to MTX treatment (GG)	G > A	A = 0.1674 (*n* = 5990)	36	46	16
rs1476413	Intron	MTHFR	Positive response to MTX treatment (CC)	C > T	T = 0.270570 (*n* = 131,256)	15	24	61
rs4149081	Intron	SLC081	Increased risk of MTX toxicity	G > A	A = 0.168487 (*n* = 136,218)	51	22	27

**Table 3 jpm-10-00149-t003:** Methotrexate Treated Cohort SNP associations. Features including plasma drug metabolite, blood cell counts and clinical outcome measures which had statistically significant associations for the four polymorphisms rs4149081, rs1476413, rs2231142 and rs17421511 in *n* = 66 methotrexate treated participants. Mean data for each genotype group with number of individuals and females per group (*n* = females/total) indicated. Statistically significant differences between genotype group means were initially assessed by ANOVA and then an adjusted by Bonferroni’s multiple comparison test performed for specified genotype group mean comparisons; asterisk used to summarise p values: (*) *p* < 0.05; (**) *p* < 0.005. Features with significant differences between genotype means are graphed in Figure 2 and Figure 3. Unlisted features had no statistically significant association with any SNP (see Appendix A). BL: baseline; 6wk: six-week follow-up; SD: standard deviation; ns: not significant.

SNP	Feature (Time pt.)	Units	Mean	±SD	Mean	±SD	Mean	±SD	ANOVA *p* Value	Genotype Comparison	Bonferroni *p* Value	Summary
**rs4149081**			GG (*n* = 25/36)		GA (*n* = 11/13)		AA (*n* = 10/17)					
	MTX (BL)	nM	28.0	7.5	35.8	7.8	28.9	8.2	0.011	GG vs. GA	**0.010**	**
	MTX (6wk)	nM	26.0	10.6	34.1	11.5	25.9	10.2	0.058	GG vs. GA	0.071	ns
	MTX-7-OH (BL)	nM	20.6	4.8	24.8	5.3	18.2	4.8	0.003	GG vs. GA	**0.038**	*
										GA vs. AA	**0.002**	**
	MTX-7-OH (6wk)	nM	19.1	6.8	23.3	7.2	17.3	7.6	0.079	GG vs. GA	0.239	ns
										GA vs. AA	0.084	ns
	Bilirubin (BL)	mg/L	6.7	2.623	8.1	6.211	11.9	10.15	0.024	GG vs. AA	**0.020**	*
	Bilirubin (6wk)	mg/L	6.6	2.392	7.8	3.76	10.2	8.64	0.059	GG vs. AA	0.054	ns
**rs1476413**			CC (*n* = 6/8)		CT (*n* = 13/16)		TT (*n* = 26/42)					
	MTX-7-OH (BL)	nM	19.8	5.6	21.2	5.7	20.1	5.9	0.791	CC vs. CT	>0.999	ns
	MTX-7-OH (6wk)	nM	12.7	6.5	22.3	5.2	18.7	8.2	0.015	CC vs. CT	**0.012**	*
	MTX 4PG (BL)	nM	19.8	5.6	37.7	16.2	32.4	15.1	0.024	CC vs. CT	**0.020**	*
	MTX 4PG (6wk)	nM	19.0	15.6	42.4	16.8	30.2	17.5	0.007	CC vs. CT	**0.008**	**
	DAS28ESR (BL)		2.3	1.5	4.0	1.6	3.6	1.5	0.047	CC vs. CT	**0.049**	*
	DAS28ESR (6wk)		3.0	1.4	4.2	1.6	4.2	1.2	0.051	CC vs. CT	0.090	ns
**rs2231142**			GG (*n* = 3/6)		GT (*n* = 12/15)		TT (*n* = 30/45)					
	RBC (BL)	mill./uL	4.4	0.4	4.1	0.3	4.4	0.4	0.050	GT vs. TT	**0.044**	*
	RBC (6wk)	mill./uL	4.4	0.3	4.1	0.3	4.4	0.4	0.042	GT vs. TT	**0.044**	*
	ALP (BL)	u/L	78.3	26.5	66.6	20.3	84.2	20.2	0.023	GT vs. TT	**0.019**	*
	ALP (6wk)	u/L	84.3	31.4	72.7	15.2	85.1	23.0	0.184	GT vs. TT	0.208	ns
	Lymphocytes (BL)	(cells/ml)	2.1	1.2	1.4	0.3	1.6	0.5	0.048	GG vs. GT	**0.043**	*
	Lymphocytes (6wk)	(cells/ml)	2.2	1.2	1.4	0.4	1.6	0.6	0.023	GG vs. GT	**0.019**	*
**rs17421511**			GG (*n* = 18/26)		GA (*n* = 24/31)		AA (*n* = 4/9)					
	PgPain (BL)	%	44.8	30.4	53.8	31.5	22.1	26.6	0.037	GA vs. AA	**0.033**	*
	PgPain (6wk)	%	45.4	27.9	55.9	32.8	22.2	23.4	0.016	GA vs. AA	**0.013**	*
	DAS28ESR (BL)		3.7	1.6	3.9	1.5	1.8	0.6	0.002	GA vs. AA	**0.002**	**
										GG vs. AA	**0.005**	**
	DAS28ESR (6wk)		4.1	1.1	4.2	1.4	3.1	1.5	0.088	GA vs. AA	0.097	ns
										GG vs. AA	0.159	ns
	ALT (BL)	u/L	20.6	10.1	21.8	11.1	32.0	15.9	0.047	GG vs. AA	**0.048**	*
	ALT (6wk)	u/L	21.7	10.7	20.5	9.3	27.1	12.4	0.252	GG vs. AA	0.534	ns

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
