# Peer review of "Clinical and Laboratory Associations with Methotrexate Metabolism Gene Polymorphisms in Rheumatoid Arthritis"

_jpm, 2020, doi:10.3390/jpm10040149_

Round 1
Reviewer 1 Report
The submitted manuscript is pleasant to read. I liked the way authors have presented and discussed the data. I have following major corrections/suggestions for otherwise nicely drafted manuscript:
- In the paragraph 3.1, it is highly confusing when the participants in the specific subgroups received which medication at what time and for how long. I insist authors to make the clear and concise table of total number of participants, subgroups, divide subgroups appropriately according to the medications.
- In the same paragraph 3.1, the gender of subgroups must be mentioned, for instance, out of 66 participants in MTX subgroup how many were males and females so on.
- Smoking status, the medications taken by participants along with MTX for other conditions affect plasma levels of MTX and its metabolites. Kindly provide these details, if not, please mention as important limitations of the study.
- In the paragraph 3.3 and so on wherever applicable, please mention gender of participants, if possible, to remove gender bias of these SNPs.
- Lines 246-247, higher or lower or variable plasma concentrations of methotrexate and sulfasalzine.......?
- Lines 336-341, liver enzymes were in the normal range with MTX, is it when participants were only on MTX or MTX + 5-ASA or MTX + Leflunomide?
- 6-weeks time point in the manuscript means 6-weeks with monotherapy or combination therapy? why 6-week time point is selected, is it to avoid heptotoxic and/or pancytopenic ADRs with MTX?
- Do any of the changes in MTX metabolism due to the studied SNPs correspond to MTX ADRs? Any available data? I read that some SNPs correlated with elevated disease activity and lower pain scores, but MTX ADRs were not mentioned.
Author Response
Reviewer 1 Comments and Suggestions for Authors
The submitted manuscript is pleasant to read. I liked the way authors have presented and discussed the data. I have following major corrections/suggestions for otherwise nicely drafted manuscript:
Response: ‘We thank the reviewer for their constructive comments and have addressed each of their points numbered below’
- In the paragraph 3.1, it is highly confusing when the participants in the specific subgroups received which medication at what time and for how long. I insist authors to make the clear and concise table of total number of participants, subgroups, divide subgroups appropriately according to the medications.
Response: ‘Thanks for raising this point, for clarity a new table (Supp. Table 2B) has been produced to indicate prescribing across the study subgroups identified in 3.1; the text at 3.1 has been simplified and the new table referred to.’
- In the same paragraph 3.1, the gender of subgroups must be mentioned, for instance, out of 66 participants in MTX subgroup how many were males and females so on.
Response: ‘Now included, thanks’
- Smoking status, the medications taken by participants along with MTX for other conditions affect plasma levels of MTX and its metabolites. Kindly provide these details, if not, please mention as important limitations of the study.
Response: ‘No cancer patients were included; smoking status was not recorded. A statement to this effect is now included in the discussion section now refers to this.’
- In the paragraph 3.3 and so on wherever applicable, please mention gender of participants, if possible, to remove gender bias of these SNPs.
Response: ‘Table 3 has been modified to indicate the number of females per genotype’
- Lines 246-247, higher or lower or variable plasma concentrations of methotrexate and sulfasalzine.......?
Response: ‘This statement draws attention to two SNPs which impact upon plasma drug concentration; the allele specific effect on concentration is discussed thereafter’
- Lines 336-341, liver enzymes were in the normal range with MTX, is it when participants were only on MTX or MTX + 5-ASA or MTX + Leflunomide?
Response: ‘This statement refers to the whole study cohort (n=100) and drug combination treated subgroups were not investigated.’
- 6-weeks time point in the manuscript means 6-weeks with monotherapy or combination therapy? why 6-week time point is selected, is it to avoid heptotoxic and/or pancytopenic ADRs with MTX?
Response: ‘6 weeks was selected as an adequate time period over which to detect disease flare ups in a patient group with active disease.’
- Do any of the changes in MTX metabolism due to the studied SNPs correspond to MTX ADRs? Any available data? I read that some SNPs correlated with elevated disease activity and lower pain scores, but MTX ADRs were not mentioned.
Response: ‘Significant ALP, ALT and blood cell count associations were observed for rs2231142 and rs17421511 SNPs, as described in section 3.4. No patient reported adverse drug reaction associations were observed.’
Reviewer 2 Report
D’Cruz et al investigated the relation(s) between 10 SNPs and the metabolite levels in 100 patients with RA and taking DMARDs. The methodology, especially the statistical analysis is sound. The plagiarism check did not show any overlap with previously published data. The study is interesting. However, some comments need to be addressed:
Major:
Line 188: The authors have conducted their study on a limited sample size (N=100) and for some SNPs (such as rs3740065) they had only one homozygous genotype. Did they try to carry out any sampling/experimental design before conducting the study? Determining the optimal sample size for their study could provide readers with an adequate number of participants to detect significant, robust results! Authors are encouraged to calculate their study's power and add this information to the statistical analysis section.
Lines 210-213: The authors cite the exact values in the text (59.5 years +/- 12.6, 3.6 (± 1.5), etc) despite being displayed in Table 1. The authors are encouraged to decrease this overlap with the Table and instead summarize these findings. Readers can retrieve this information from the Table.
Line 238: Table 2 would fit more in the supplementary data… However, the columns SNP and genotype frequency (please correct, not allele frequency) can be added to Table 1 instead of after the line 201 (Bilirubin mg/L).
Line 239: The SNP rs4846051 in MTHFR did not show any presence of the minor allele, the 100 individuals were homozygous WT (it has a low MAF in public databases (0.02)), the authors can add this variant to supplementary data.
Line 394: the current study lacks a limitation section, authors could discuss the limited number of individuals being investigated.
Minor:
Symbols for the genes need to be italicized all over the manuscript (abstract, figures also), whereas for proteins, the symbols are regular.
Lines 28, 29, and 31: The authors can remove the term inside the brackets (C.I. 95%).
Line 98: The authors need to avoid abbreviations in the titles: such as DBS.
Line 332: in vivo needs to be written in italic.
Author Response
Reviewer 2 Comments and Suggestions for Authors
D’Cruz et al investigated the relation(s) between 10 SNPs and the metabolite levels in 100 patients with RA and taking DMARDs. The methodology, especially the statistical analysis is sound. The plagiarism check did not show any overlap with previously published data. The study is interesting. However, some comments need to be addressed:
Response: ‘We thank the reviewer for their constructive comments and have addressed each of their points numbered below’
Major:
- Line 188: The authors have conducted their study on a limited sample size (N=100) and for some SNPs (such as rs3740065) they had only one homozygous genotype. Did they try to carry out any sampling/experimental design before conducting the study? Determining the optimal sample size for their study could provide readers with an adequate number of participants to detect significant, robust results! Authors are encouraged to calculate their study's power and add this information to the statistical analysis section.
Response: ‘We have assessed if there were any significant differences in Hardy-Weinberg equilibrium between the genotype frequencies expected from the dbGAP European population and those observed in the methotrexate treated cohort within the study. We found no significant differences by Chi-square test followed by Benjamini-Hochberg correction of p-values. Please refer to our updated methods and results section. Therefore, even though our cohort size is small but the genotype frequencies are in-line with the European population, meaning the findings are generalizable to the European population. Additionally a power calculation for each of these four SNPs with statistically significant associations has now been made and inserted into the results section’
- Lines 210-213: The authors cite the exact values in the text (59.5 years +/- 12.6, 3.6 (± 1.5), etc) despite being displayed in Table 1. The authors are encouraged to decrease this overlap with the Table and instead summarize these findings. Readers can retrieve this information from the Table.
Response: ‘Text has been amended to address this comment, thanks’
- Line 238: Table 2 would fit more in the supplementary data… However, the columns SNP and genotype frequency (please correct, not allele frequency) can be added to Table 1 instead of after the line 201 (Bilirubin mg/L).
Response: ‘Thanks for the suggestion, Table 2 has now been corrected. For clarity the genotype results of the study are kept separate from the participant clinical and demographic information.’
- Line 239: The SNP rs4846051 in MTHFR did not show any presence of the minor allele, the 100 individuals were homozygous WT (it has a low MAF in public databases (0.02)), the authors can add this variant to supplementary data.
Response: ‘As above, the genotype frequency of all SNPs investigated are shown together for clarity.’
- Line 394: the current study lacks a limitation section, authors could discuss the limited number of individuals being investigated.
Response: ‘A statement on the number of participants as a limitation of the study has been inserted in to the discussion.’
Minor:
- Symbols for the genes need to be italicized all over the manuscript (abstract, figures also), whereas for proteins, the symbols are regular.
Response: ‘Thanks, the gene symbols have been formatted throughout.’
- Lines 28, 29, and 31: The authors can remove the term inside the brackets (C.I. 95%).
Response: ‘now removed thanks.’
- Line 98: The authors need to avoid abbreviations in the titles: such as DBS.
Response: ‘Now removed; SNP and DNA have also been replaced in subtitles.’
- Line 332: in vivo needs to be written in italic.
Response: ‘Now corrected, thanks.’
Round 2
Reviewer 1 Report
The authors responded appropriately to all of my concerns.
However, for point no 6, if authors could kindly include in the limitations that liver enzymes were not studied in the subgroup of drug combinations studied and only measured for MTX treatment, it would be accurate for the publication.
Apart from that, all the corrections/additions/comments were acceptable. I am really looking forward to see this manuscript published.
Congrats!
Author Response
However, for point no 6, if authors could kindly include in the limitations that liver enzymes were not studied in the subgroup of drug combinations studied and only measured for MTX treatment, it would be accurate for the publication.
Response: ’ A sentence in the discussion section has been inserted for clarity: ‘The mean values of all clinical biomarkers, liver enzyme and blood component cell-counts (Table 1) are within recommended normal reference ranges when viewed across all of the study participants. However, mean ALT was significantly higher at baseline in the methotrexate treated subgroup of participants with the rs17421511 AA genotype, albeit the potential effect of multiple drug combinations was not investigated in this subgroup.’'
Apart from that, all the corrections/additions/comments were acceptable. I am really looking forward to see this manuscript published.
Response: ‘Many thanks for your constructive feedback’
Reviewer 2 Report
The authors have answered my comments
Author Response
Response: ‘Many thanks for your constructive feedback’